# Identifying and Classifying Urban Data Sources for Machine Learning-Based Sustainable Urban Planning and Decision Support Systems Development

Stéphane C. K. Tékouabou [1,*] , Jérôme Chenal [1,2] , Rida Azmi [1] , Hamza Toulni [3,4] and El Bachir Diop [1] and Anastasija Nikiforova [5,6]

1 Center of Urban Systems (CUS), Mohammed VI Polytechnic University (UM6P), Hay Moulay Rachid, Ben Guerir 43150, Morocco
2 Urban and Regional Planning Community (CEAT), Ecole Polytechnique Federale de Lausanne (EPFL), 1015 Lausanne, Switzerland
3 EIGSI, 282 Route of the Oasis, Mâarif, Casablanca 20140, Morocco
4 LISTD Laboratory, Department of Computer Sciences, Mines School of Rabat, Av Hadj Ahmed Cherkaoui, Agdal, P.O. 753, Rabat 10090, Morocco
5 Institute of Computer Science, Faculty of Science and Technology, University of Tartu, Narva mnt 18, 51009 Tartu, Estonia
6 European Open Science Cloud Task Force "FAIR Metrics and Data Quality", 1050 Brussels, Belgium
* Correspondence: stephane.koumetio@um6p.ma or ctekouaboukoumetio@gmail.com; Tel.: +212-771-515-963

**Abstract:** With the increase in the amount and variety of data that are constantly produced, collected, and exchanged between systems, the efficiency and accuracy of solutions/services that use data as input may suffer if an inappropriate or inaccurate technique, method, or tool is chosen to deal with them. This paper presents a global overview of urban data sources and structures used to train machine learning (ML) algorithms integrated into urban planning decision support systems (DSS). It contributes to a common understanding of choosing the right urban data for a given urban planning issue, i.e., their type, source and structure, for more efficient use in training ML models. For the purpose of this study, we conduct a systematic literature review (SLR) of all relevant peer-reviewed studies available in the Scopus database. More precisely, 248 papers were found to be relevant with their further analysis using a text-mining approach to determine (a) the main urban data sources used for ML modeling, (b) the most popular approaches used in relevant urban planning and urban problem-solving studies and their relationship to the type of data source used, and (c) the problems commonly encountered in their use. After classifying them, we identified the strengths and weaknesses of data sources depending on several predefined factors. We found that the data mainly come from two main categories of sources, namely (1) sensors and (2) statistical surveys, including social network data. They can be classified as (a) opportunistic or (b) non-opportunistic depending on the process of data acquisition, collection, and storage. Data sources are closely correlated with their structure and potential urban planning issues to be addressed. Almost all urban data have an indexed structure and, in particular, either attribute tables for statistical survey data and data from simple sensors (e.g., climate and pollution sensors) or vectors, mostly obtained from satellite images after large-scale spatio-temporal analysis. The paper also provides a discussion of the potential opportunities, emerging issues, and challenges that urban data sources face and should overcome to better catalyze intelligent/smart planning. This should contribute to the general understanding of the data, their sources and the challenges to be faced and overcome by those seeking data and integrating them into smart applications and urban-planning processes.

**Keywords:** urban data source; urban sensing; remote sensing; data structure; opportunistic data; machine learning; artificial intelligence; urban planning; text mining

## 1. Introduction

Data are a central object to informed decision making, even more so for more advanced solutions, such as business intelligence, models, forecasts, prediction, planning, and decision support systems (DSS), which are significantly affected by data as input on which they are based or trained. These data and the corresponding data source(s) must be consistent with the purpose of the project. A project that models and/or monitors the performance of relevant indicators for a given urban issue, such as urban growth [1], land use/coverage [2,3], housing and slums [4], mobility [5,6], climate [7,8], building [9], pollution and air quality [10,11], energy [12,13], environment [14], waste management [15], etc., by using techniques such as machine learning (ML) usually begins with the acquisition or collection of the data for the desired area of the study [16].

Despite significant advances in data science, many initiatives start and stop at this stage because researchers either do not know where to find, how to collect, or how to use the often unstructured and highly complex data found that are now increasingly often freely available, sometimes in the form of open data. Artificial intelligence (AI) based on ML is currently relevant in urban planning and urban sustainability, planning and developing Smart Cities, including but not limited to CityLab initiatives [17] where data acquisition is the raw material and starting point [16], making understanding the data crucial.

Moreover, the emergence of various technological advances, such as the internet of things (IoT), smart technologies and smart city technologies in particular [18], digital twins, internet 2.0, widely available, accessible and used smart devices by almost all citizens, make the city a huge data provider/producer [19]. In addition, rapid improvements in machine computing capacity, and the availability of real-time or almost real-time data lead to revolutionary advances in the simulation and modeling of complex systems [16]. A city described in this way can be a typical example of a complex system, with its dynamic network of interacting and constantly changing human, institutional, environmental and physical systems [13]. However, very little research has been conducted so far to highlight or attempt to categorize sources of urban data in order to better guide and accompany this new research dynamic. This study makes such an attempt.

In this paper, we discuss urban data sources and structures related to applications of ML methods for modeling and monitoring urban form indicators in planning, including the data related to intelligent urban planning decision support systems and tools. To this end, we use the research methodology presented in [16] to better circumscribe relevant literature. More precisely, we conduct a systematic literature review (SLR) of all relevant peer-reviewed studies available in the Scopus database. In total, 248 papers found to be relevant are analyzed, using text-mining techniques to determine (1) the main urban data sources used for ML modeling. (2) What are the most popular approaches used in relevant urban planning and urban problem-solving studies and their relationship to the type of data source used? (3) What are the problems commonly encountered in their use? After classifying them, we also identify the strengths and weaknesses of data sources depending on device type, referring to complexity, spatial coverage, temporal information, open accessibility, and cost.

The current research suggests that it is difficult to find an exhaustive classification/categorization of urban data sources, which are currently high and increasingly divergent [16]. This categorization varies from source to source, depending on the field of study and context. Some authors talk about opportunistic and non-opportunistic data, which are based on the means of network communication [20]), or even opportunistic or participatory data, depending on whether the participants are known and notified in advance [21]. However, this initially very logical categorization (based on the protocol for collecting data over the internet) has now been adapted to physical resources, or both in some cases [22]. Other authors discuss vectorized (spatial) or attribute data (data structure-based categorization) [23]. Some experts simply discuss sensor data, defining them as a single type of urban data, i.e., the data that come from dedicated sensors [23,24]. However, what is the primary source of this data? The word "source" refers to the "origin", or "place

where something comes from". Thus, data are first collected (captured/detected) from somewhere for a well-defined geographical area (called the "study area") using a fixed or mobile onboard currency and then sent/routed in real-/delayed-time or otherwise for their further processing and storage [24]. This process is illustrated in Figure 1, which allows for the identification of the levels that each study is at for categorizing urban data sources. Except for sensor data, statistical surveys are another and well-known (old) important source of urban data for extracting and further exploring information on various indicators [25]. The information describing the data from these two sources allows the generation of urban metadata, which is often useful, especially for the reuse of data [26]. Whether these urban data come from sensors or statistical surveys, they are stored and shared under open access or on licensed platforms that are often assimilated to their source.

This is expected to provide the reader with an understanding of where data for the desired urban planning problem can be found and what are the main sources of urban data for ML applications in urban planning.

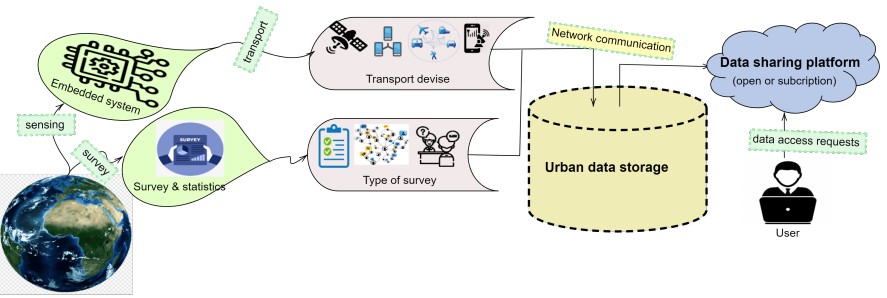

**Figure 1.** Flowchart (or workflow) of the urban data collection process.

Hence, the rest of the paper is structured as follows: Section 2 discusses spatialized data, GIS being one of the most crucial and popular urban data sources. Section 3 presents the research methodology, while the analysis of the results, discussion and conclusion are then provided in Sections 4 and 5, respectively.

## 2. Urban Data

First, let us elaborate on "urban data" as the central object of this study. In this paper, we understand urban data as numerical values obtained from measurements and other sources in the urban environment [25]. According to the current state of the art, these data can and tend to be made open, i.e., they can be freely accessed (online) and reused without any technical or legal restrictions [25,27,28], which makes them more valuable. The importance of the openness of the data that could be used for ML-based applications have become more expressed today, and there are many specific platforms, such as Kaagle, providing the user with the datasets specifically for this purpose.

Data sources are generally classified as (1) the conventional standing for the most traditional form of collecting data from one or more sources, typically involving the searching/finding, obtaining/acquisition, assessing, integrating, and using the data; (2) crowd-based (or crowd-sourced), defined as the result of collective contribution of the general public (standing for the "crowd" in its title) to the generation, and in some cases further stages, such as aggregation, and processing of the data for further use; and (3) cloud-based, whereas the title suggests that the data are sourced from cloud platforms provided through dedicated interfaces [29].

However, in the context of urban data, data sources are typically divided into sensor networks and statistical surveys, including official statistics and, more recently, user-generated content [25]. These sources typically generate raw data that are then structured in well-defined formats before they are released to the public, i.e., shared. Platforms for accessing urban data can be mostly open source and freely accessible or under a paid license.

The diagram in Figure 1 shows the process of obtaining these data. There are two main sources of urban data—sensors and surveys (we will cover them in more detail in Section 4.2). Sensor data differ mainly by the type of devices or even type of transport collecting them, e.g., satellite, aerial, ground, ubiquitous, and fixed, while statistical survey data differ according to their type, e.g., interviews, institutional statistics and social media. These data are stored in urban data storage databases (urban databases), which are accessible by different users through data-exchange or data-sharing platforms. Let us discuss the key elements of understanding urban data and its sources.

*2.1. Data Structures*

In general, a data structure is a specific format for storing, accessing, and processing data to meet specific requirements. The data structure is typically either finite (i.e., constants, variables, point records or indexed, e.g., arrays, associative tables, and vectors) or recursive, (e.g., lists, trees, and graphs). There are many forms of structures, and their complexity varies, some of which are related to the sources and the context in which they are used. The structure determines the methodology to be used, when working with them, e.g., supervised or unsupervised approaches for structured data, their complexity, and the processing tools. An inappropriate data structure or method used to work with them can lead to slow execution time, poor performance, or complicated and unresponsive code. In other words, when choosing a data structure, it is important to consider the type of data that will be stored, the location of existing data, and how the data will be accessed and processed.

Most often, raw data are unstructured or semi-structured, and can be described by several structural parameters that indicate whether they are any of the following:

- **Linear or non-linear** indicates whether the data items are organized chronologically, as in a table, or non-graphically, as in a graph. The data can also be periodic or seasonal.
- **Homogeneous or heterogeneous**, indicating whether all data elements in a specific repository are of the same type for homogeneous data or of different types for heterogeneous, respectively. Heterogeneous data can also come from multiple sources and be aggregated or merged to better target a given indicator.
- **Static or dynamic**, describing how data structures are compiled. Static structures have fixed sizes, structures and memory locations at compile time. In a dynamic data structure, the size, structures, and memory locations can shrink or grow depending on the use of the data structure.

The triangle in Figure 2 makes a correspondence between the type of urban data structure and the quality of urban information expressed. This correspondence is further highlighted in Table 1. According to this figure, urban data can combine attribute, spatial and/or temporal variables. The latter allows respectively the question what? where? and when? to be answered based on the context-conducted studies.

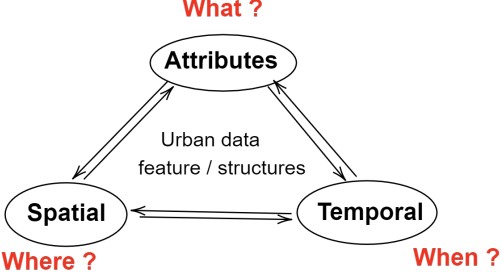

**Figure 2.** Urban data features and corresponding structures.

**Table 1.** Comparative summary of urban data structures.

| Data Struct | Questions | Description | Advantages | Disadvantages |
|---|---|---|---|---|
| Vector data | WHEN & WHERE | Made up of a grid of pixels. Instead, vector graphics are comprised of vertices and paths. The three basic symbol types for vector data are points, lines, and polygons (areas) | +Compact data structure<br>+Efficient for encoding topology<br>+True representation of shape | -Complex structure<br>-Overlay operations difficult<br>-Might imply a false sense of accuracy |
| Raster data | WHEN and WHERE | The simplest form consists of a matrix of cells (or pixels) organized in rows and columns (grid) in which each cell contains a value representing information | +Suitable for complex analysis<br>+Efficient for overlays<br>+Common for imagery where matrices are easy to analyze | -Large datasets which requires a lot of resources for processing and storage<br>-Topology hard to represent<br>-Maps are less "realistic" due to spatial resolution<br>-Difficult to adequately represent linear features depending on the cell resolution. |
| Attribute data | WHEN and WHERE and WHAT | Alphanumeric variables describing a given urban entity that may not have a spatial component (longitude and latitude). Technically, they are considered non-spatial tables that can be browsed and modified using the attribute table view in urban data analysis tools such as QGIS. | +Simple structure<br>+Suitable for Simple analysis<br>+Efficient for overlays<br>+Easy to analyze linear features<br>+Require low resources and computing expertise<br>+Low data preprocessing | -Inefficient for complex analysis<br>-Subject to appraiser interpretation<br>-Hard to represent the topology |

## 2.2. Spatialized Data Structuring and the Emergence of GIS

In recent years, geographic information systems (GIS) tools have become increasingly popular, as their hardware has become more affordable and software more user-friendly. The benefits of using GIS data in urban planning are numerous, the most important of which are the following:

▶ GIS can increase the relevance and currency of a map by providing a single source of current and historical data and maps. This increases the efficiency of thematic mapping and reduces the cost of data storage. Because GIS solutions are available in a variety of architectures—desktop or cloud—modern solutions enable improved access to vital data. Desktop GIS makes it easy to store, organize, and retrieve data from various sources, while cloud GIS does the same but from any device. GIS technologies provide improved communication through a single data storage and management system.

▶ Improved support and assistance for strategic decision-making with faster access to more relevant and up-to-date geographic data. As a consequence, planners can make informed decisions and plan more effectively. Furthermore, they can explore a wider range of what-if possibilities, resulting in more reliable, robust and successful long-term plans. All of this is possible because of spatial data that allow users to think more spatially. The context of a GIS is critical to better understanding the structuring of urban data. GIS databases are structured to ensure links between the physical objects stored in the simulation and the spatial/attribute databases so that dynamic queries can be run during the simulation [30]. In the context of GIS, a typical project involves datasets accessible at different levels of spatial aggregation: census data available by area or block; data associated with a particular site/parcel, such as zoning characteristics, estimated value, number of parking spaces; and individual building-level data, which may include building value, the condition and number of floors, and the number of employees [30–32]. These types of data attributes are typically associated with polygon features in a two-dimensional (2D) GIS database. In addition, the data are associated with linear features, such as street or utility networks (e.g., traffic or information flows [6]).

All in all, urban data usually combine (1) attribute, (2) spatial and/or (3) temporal variables, allowing to answer what, where, and when questions based on the context of the conducted studies (see Figure 2 and Table 1); however, elaborating on this in more detail, establishing a correspondence between the urban data structure type and the quality of expressed urban information, when combining the above "questions", provides a brief insight into the general description of these data, and the advantages and disadvantages that the user can face when working with them.

Generally, spatial vectors, attribute features, and metadata are the three forms of data in every spatial database. There are also several types of spatial data: (1) spatial vector data (usually points, lines, and polygons), and (2) raster data, also known as data grid, and finally (3) picture data, such as data from remote sensing (a more advanced idea).

Attribute data are indexed data, the size of which often consists of $n$ items (samples and records) in rows, described by $j$ attributes (features) in columns. Each row corresponds to an item, and the columns carry the features (characteristics) of the entities. This classic type of data can come from multiple sources based on sensors (temperature, pollution, humidity, etc.) as well as surveys. Attribute data are less complex and easier to analyze due to superior model accuracy using simple learning algorithms or statistical tools [16]. However, they have the disadvantages of being difficult or even expensive to obtain over a large geographic area [25]. Therefore, they allow work to be carried out on a small scale but with greater precision/accuracy when multiple data sets of this type are often then aggregated to obtain information on a larger scale. In addition to these two categories of data, metadata are the most neglected type of data, although they are commonly used, especially if someone else is going to use the database afterwards. They contain information about the scale, the accuracy, the projection and/or datum, and how the data were obtained/collected/generated [26].

## 3. Research Method

To create a knowledge base on a topic and identify all relevant literature to be used as a source of information for further analysis, we first performed a literature search on the Scopus database, which is considered the most comprehensive overview of the world's research findings in the technology, medicine, social science, and arts and humanities, covering over 5000 international publishers including but not limited to Elsevier, Springer, Wiley-Blackwell, Taylor & Francis, Sage, Wolters Kluwer, Oxford University Press, Emerald, Inderscience Publishers, Cambridge University Press, Bentham Science, IEEE (https://www.elsevier.com/__data/assets/pdf_file/0007/69451/Scopus_ContentCoverage_Guide_WEB.pdf, accessed on 15 January 2022). The search query combined two sets of keywords: (1) "urban planning", "urban form", "urban shape" and "urban morphology", and (2) "machine learning" and "deep learning". In the search query, the logical Boolean operators *"AND"* and *"OR"* were used to combine the keywords of the two categories and include the keywords of each, and finally *"EXCLUDE"* to remove the review, a review of books, letters or documents in a language other than English constituting a set of exclusion criteria. In other words, we searched all relevant peer-reviewed literature in English, focusing on journal articles, conference proceedings, books, chapters, etc.

This allowed us to retrieve 751 raw articles and 723 after the removal of reviews, conference reviews, and letters. Key data about the selected papers were downloaded as .csv files and then passed through a filtering process to leave only the most relevant, i.e., articles addressing the topic of machine learning applications in urban planning. In order to obtain targeted documents that refer to (or at least mention) the description and therefore the source of the urban data involved, we adapted the search elements after several filtering based on data analysis techniques using the ORANGE tool [33]. Several related articles deemed relevant and not preserved from the previous steps were manually added.

Thus, we reached 248 of the most relevant articles that were included in the analysis, leading to the results we present in the next sections.

## 4. Analysis of Results

### 4.1. Bibliometric Analysis Summary

Analysis of the results of a bibliographic survey is important for clarifying the results of the conducted survey to readers. The latter usually combines statistical analysis of publication trends, type of publications, authors, themes, sources, publishers, etc. [19]. In order to make it more insightful, we added an exploratory analysis of some key indicators to measure and show the reliability of the research conducted using indicators from the two categories. The first is related to scientific publications, namely the number of publications (Figure 3a), authors' affiliating institutions (Figure 4a), keywords (Figure 5), source journals (Figure 3b), and access to/types of papers, while the second refers to scientific impact expressed as the citations count.

Our analysis shows that there is a trend toward a strong increase in the number of publications on this topic that are indexed in Scopus database (see Figure 3a). Starting with one article published per year before 2012, this number grows exponentially from this year until it reaches 71 in 2021. As for the top journals, *IEEE access* and *Remote Sensing* source journals from IEEE and MDPI publishers, Open Access is the most popular (see Figure 3b). Other sources on this list are Elsevier (the leader of the list based on the number of journals), Springer, and SPIE.

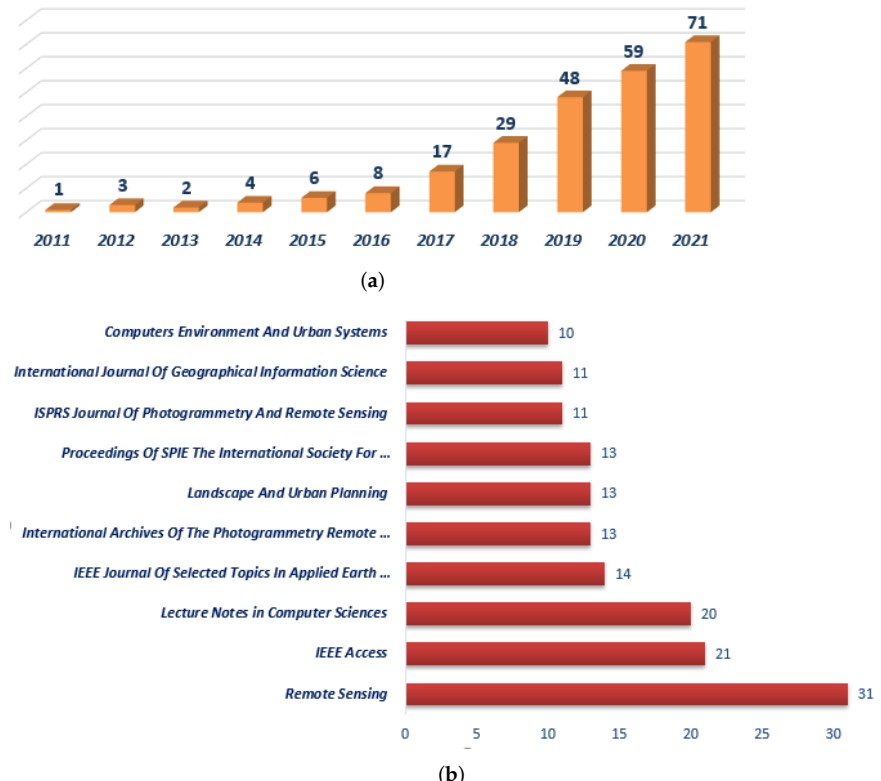

**Figure 3.** Number of documents published by (**a**) year, (**b**) source journal.

Figure 4a,b show the top 10 institutions in terms of the number of published articles and the relationship between the most frequent institutions' countries, respectively. As with many current scientific research topics, this list is dominated by Chinese institutions and a few American and European institutions. The dominance of Chinese institutions in this list can be justified not only by their intense ongoing research activities, but also by the dense networks to increase the number of publications. However, completing this analysis as a whole, the US, then Chinese, and UK institutions are the largest contributors.

Another point we analyzed was the type of and access to scientific publications, which is found to be important to assess their quality and scientific impact. A total of 43% articles are available in open access, where gold and green open access are found to be the most popular routes, i.e., 28% and 22%, respectively. For the type of these contributions, the majority, i.e., 57% of documents are journal articles, while conference papers and book chapters represent 34% and 9%, respectively. Knowing that journal articles not only have a relatively longer publication process and stricter evaluation criteria, their dominance shows not only the reliability of the research carried out, but also the maturity of the research topic studied.

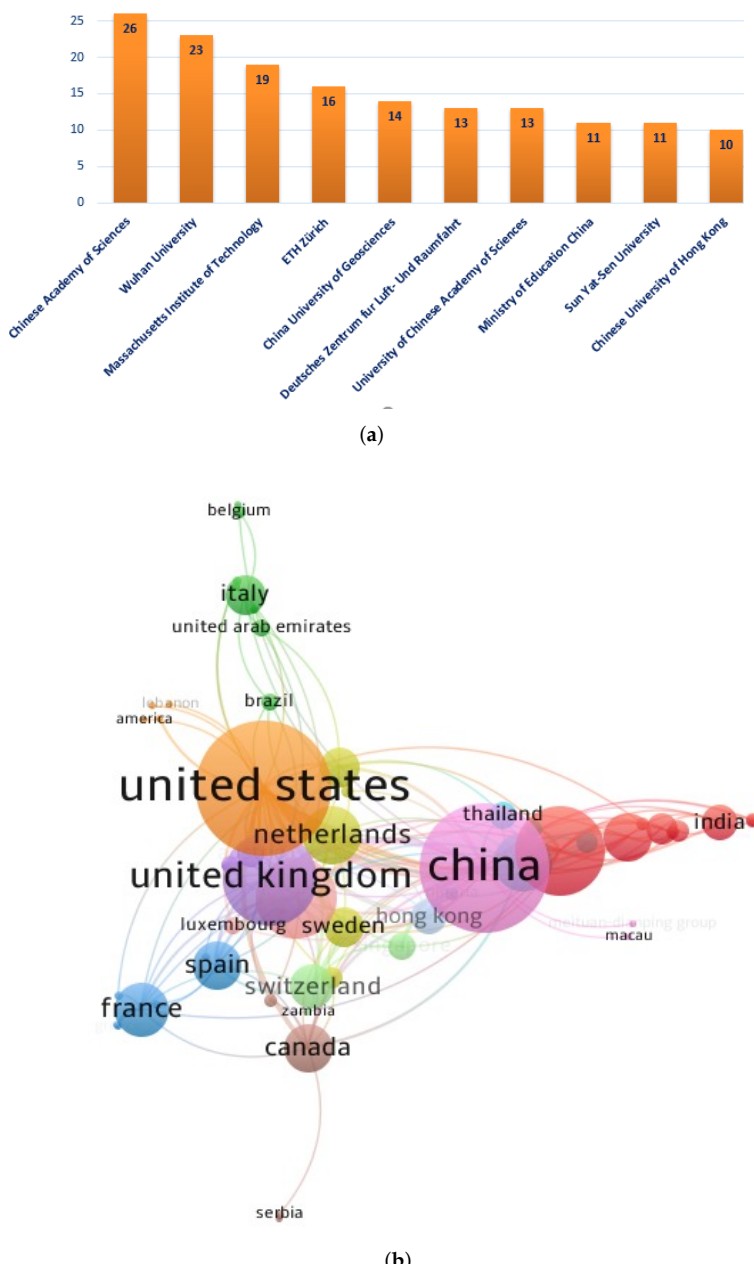

(a)

(b)

**Figure 4.** (**a**) Top 10 affiliated institutions and (**b**) interconnections between the most frequent institutions' countries.

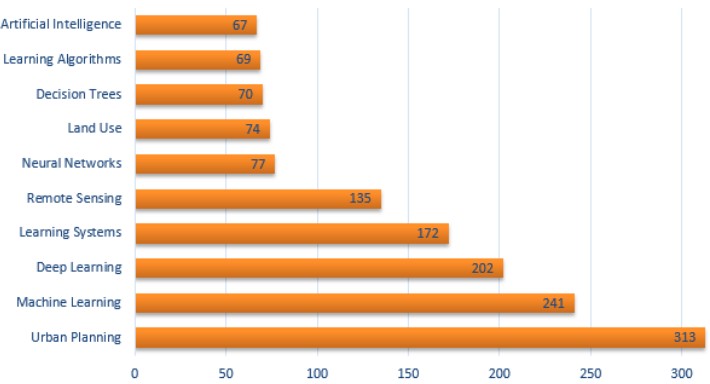

**Figure 5.** Relations between the most frequent countries.

As regards the scientific impact measured as the number of citations that the contributions received, we found that several papers on the issues faced during the application of ML methods in urban planning were widely disseminated and cited. The top ten most cited articles in the Scopus database with their brief summary are provided in Table 2. For the summary we provide, being in line with the purpose of this study, we focus on the data sources used, as there is a close relationship between the data source, target indicators and ML methods used. Regarding scientific impact, we consider the number of citations of these articles, which ranges from 288 for the most cited to 71 for the least cited. We thus calculate the average number of citations per year (ACPY), which varies depending on the year of publication. It ranges from 11 for the lowest to 67 for the highest. The publishers in this top ten are Elsevier with 5 articles, followed by IEEE with 4 articles and finally Taylor & Francis with one article. For the data sources used in these studies, eight sources come exclusively from sensors, more specifically from satellites and telecommunications devices, while one is based on hybrid data (sensor and survey) and only one is based on survey data.

**Table 2.** Top 10 most cited papers. ACPY = Average Cites Per Year.

| N° | Ref | Year | Data Source | Source | Cites | ACPY | Publisher |
|---|---|---|---|---|---|---|---|
| 1 | [34] | 2012 | Sensing (satellite) | Remote Sensing of Environment | 288 | 32 | Elsevier |
| 2 | [35] | 2018 | Sensing (satellite) | Remote Sensing of Environment | 202 | 67.33 | Elsevier |
| 3 | [36] | 2015 | Sensing (satellite) | International Geoscience and Remote Sensing Symposium (IGARSS) | 199 | 33.170 | IEEE |
| 4 | [37] | 2017 | Survey | Applied Energy | 155 | 38.75 | Elsevier |
| 5 | [38] | 2018 | Hybrid | Landscape and Urban Planning | 103 | 34.33 | Elsevier |
| 6 | [39] | 2012 | Sensing (OpenStreetMap) | International Journal of Geographical Information Science | 92 | 10.22 | Tay & Fr |
| 7 | [40] | 2013 | Survey (Telecom & Geotagué des phones) | Proceedings—IEEE International Conference on Mobile Data Management | 91 | 11.36 | IEEE |
| 8 | [14] | 2017 | Sensing (Baidu Map) | Computers, Environment and Urban Systems | 88 | 22 | Elsevier |
| 9 | [41] | 2017 | Sensing (satellite) | IEEE Geoscience and Remote Sensing Letters | 78 | 19.5 | IEEE |
| 10 | [42] | 2017 | Sensing (GPS trajectory dataset) | IEEE Transactions on Knowledge and Data Engineering | 71 | 17.75 | IEEE |

Going a step further, we analyzed the most frequently used keywords. Figure 5 shows the ten most frequently used keywords, confirming some of those included in the search query, i.e., "urban planning", "machine learning" and "deep learning" at the top of the list. Then referring to [16], we categorize the other keywords of this shortlist into (a) "learning systems", (b) "neural networks", (c) "decision trees", (d) "learning algorithms" and (e) "artificial intelligence" associated with machine learning, followed by (f) "land use", which is the most commonly modeled urban planning indicator in published research, and last but not least, (g) "remote sensing", which is the most commonly used urban data acquisition technique to train ML methods, in this case to model indicators such as "land use".

Now, let us refer to the key research object of this study, i.e., sources of urban data analysis considered in the literature.

### 4.2. Sources of Urban Data Analysis

In order to better adapt to the defined task, ML algorithms should receive (be fed with) the "best" data [43]. In the field of urban planning, these data come from various sources depending on the questions to be addressed and targeted indicators [16]. Researchers have relied on a variety of data sources that often correlate with the issues under study, ranging from the most classic in urban science (sensing and surveys) to the most recent and emerging, such as ubiquitous data [25]. The emergence of urban data capable of training powerful ML algorithms has enabled the provision of more intelligent solutions to current challenges of cities, such as "sustainable planning, smart city, digitalization,

resilience, developing better strategies and studying the impact of new urban development projects" [16]. Generally speaking, urban data sources are sensor-based technologies and data from institutional surveys [25].

Our analysis of the current literature shows that (1) 58% studies applying ML use data coming exclusively from sensors, (2) 33% use hybrid data, and finally (3) 9% use surveys exclusively (see Figure 6). These data sources give an idea of their reliability according to the desired study and the real impact of the results obtained.

Let us elaborate on these data sources in more detail.

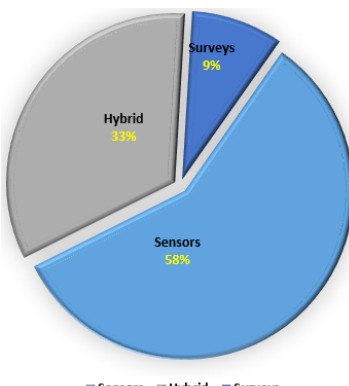

**Figure 6.** Distribution of urban data sources.

### 4.3. Data Sources: Remote Sensing and Surveys

#### 4.3.1. Remote Sensing

Sensing is the most important step in obtaining most urban data [8,24]. This is performed using onboard sensors that are fixed somewhere on the device to cover an area of a smaller spatial scale or carried in a mobile device [44], whose speed depends on the area covered and the accuracy of the collected data. Surveys conducted show that more than 58% of urban data used to train the ML algorithms comes from sensors, while 33% are hybrid sensing and survey data [16].

Nowadays, high volumes of highly diversified sensor data are generated, and this volume is expected to grow exponentially [24]. Most often, they are geographically far from their target, which is why this technique is called "*remote sensing*", which is the most commonly used for collecting urban data. Considering the sources of sensor-based data, it is possible to classify/categorize them depending on the transport technologies used, which reflect the spatial extent of the area covered and the scale of data precision/accuracy [16].

We distinguish several types of data depending on the type of carrier device: (1) satellite/radars, (2) drone/plane, (3) ground transports such as bikes, cars, scooters and e-scooters, trains, (4) ubiquitous mobile devices, (5) fixed devices, (6) social networks and social media, (7) crowd-sourcing, (8) interviews, and (9) institutional statistics. Table 3 presents their description as well as provides an insight into their advantages and disadvantages, assessing their (a) complexity, (b) spatial coverage, (c) temporal information, (d) accessibility in open access, (e) cost, and data structure, i.e., (a) vector, or (b) attribute. Complexity and cost are assessed as either low or high spatial coverage (as global or partial), while temporal information and accessibility as a Boolean function (yes or no).

**Table 3.** Comparative summary of urban data sources for ML modeling.

| DS | Device Type | Description | Advantages & Disadvantages | | | | | Data Structure | |
|---|---|---|---|---|---|---|---|---|---|
| | | | Complexity | Spatial coverage | Temporal infos | Open Accessibility | Cost | Vectors | Attribute |
| - | | | | | | | | | |
| Sensors | Satellite/radars | Data generated by remote sensing technologies using sensors carried by satellites | Low | Global | YES | Yes/No | Low | ✓ | |
| | Drône/plane | Data from sensors built into air vehicles (planes, drones, helicopters, etc.) | Low | partial | YES | No | High | ✓ | ✓ |
| | Bikes/Cars/ Motos | Data from sensors on or integrated into ground transporters (bicycles, motorbikes, buses, taxis, trains, etc.) | Low | Partial | YES | Yes | Low | | ✓ |
| | Ubiquitous mobile devices | Data from sensors embedded in any mobile device or any other connected device of daily use (phones, watches, smart homes/infrastructures, etc) | Low | Partial | YES | Yes | Low | | ✓ |
| | Fixed devices | Data from sensors either embedded in a dedicated fixed device or in any other device opportunistically (e.g., camera, street lights, ...) | High | Partial | YES | No | High | ✓ | ✓ |
| Survey and institutional Statistics | Social Networks/media | Emerging data from the social networks (including web surveys) such as Facebook, Twitter, etc. | High | Global | YES | Yes | Low | | ✓ |
| | Crowd-sourcing | Data from a large group of people in a study area, who submit (voluntarily) their data via the internet, social media, or smartphone applications | High | Partial | No | No | Low | ✓ | ✓ |
| | Interviews | Data from interviews on urban issues that can be conducted offline or online via social media | High | Partial | NO | No | High | | ✓ |
| | Institutional statistics | Data from governmental and non-governmental institutions' statistics | Low | Partial | NO | No | High | ✓ | ✓ |

### 4.3.2. Survey/Statistical Urban Data

In addition to the sensor-based data discussed earlier, survey data are one of the main and oldest sources of urban data. These data come from many federal, state, regional, and local government agencies, including, but not limited to national open government data portals, but also non-governmental institutions, as well as various private sources [28]. They also include data repositories for ML, such as the UCI machine learning repository, Kaggle, KDD, etc. While the opportunity to re-use data is very beneficial and allows to reduce the number of resources to be spent (both human, time, and fiscal), for many research tasks, and even more so for training ML algorithms, relevant data are not available or even do not exist, thus making it necessary for urban researchers to collect these data themselves [25]. This becomes even more vital if up-to-date data characterized as being rather dynamic in nature are the object of interest, i.e., where the data available under some conditions could become outdated and useless or insufficient.

While surveys are the primary method for collecting new data, urban-planning researchers also use a number of more creative data generation methods. These methods increasingly incorporate ICT and social participation [19]. Individual or grouped citizen

initiatives play an important role in the collection and provision of such data. Therefore, we classify this category of data as (a) data from institutional (statistical) surveys, (b) social networks (also sometimes referred to as web sensing), (c) interviews, (d) crowdsourcing [45], etc. These data are often used alone [15] or in combination with sensor data, thereby increasing the level of completeness and reliability [31]. In addition, they are more opportunistic and easy to collect and process than sensing data (see Table 3).

There is also evidence found in the literature reporting on the cases when authors combined survey data and sensor data to address a socio-functional or socio-technical aspect of the city with an urban form indicator [16]. For instance, [46] addressed the correlation of the physical activity of city users (obtained by survey) to the greenness of the streets (obtained by Google street view data).

### 4.4. ML Methods Depending on the Urban Data Source

The decision to use ML methods is heavily influenced by data-related factors, such as sources and structures. When faced with an ML problem, one of the most commonly asked questions is "Whether it is supervised learning (the target is known in advance) or otherwise unsupervised". Whenever the target is known for supervised learning, it can be discretely or nominally associated with a classification problem; otherwise, it will be nominal and therefore associated with a regression problem [16].

The conducted analysis of the existing body of knowledge shows that the use of supervised learning methods largely dominates over unsupervised learning methods, regardless of the data source used (Figure 7). This is not only because in most cases, predictive modeling data naturally have a target variable known in advance, but can also be obtained in an intermediate step of unsupervised learning, such as clustering.

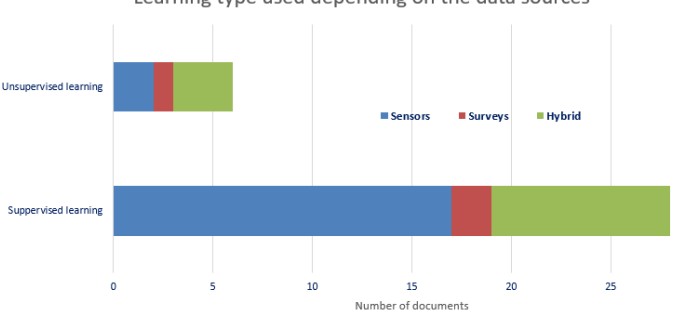

**Figure 7.** Learning type depending on the data sources involved.

There is also the dominance of regression problems over classification problems for supervised learning, and the dominance of association rules over clustering for unsupervised learning shown in Figure 8. However, analysis related to data sources reveals that for sensor data (mostly satellite images), the learning problem is mainly in classification rather than regression. The opposite observation is made for hybrid data or data obtained exclusively from surveys.

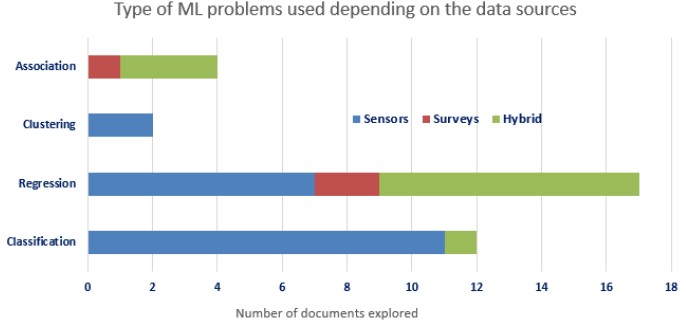

**Figure 8.** Learning problems used depending on the data sources involved.

Now it remains to find methods that are most commonly used according to these different types of problems. To do this, it is necessary to address the ML and the DL separately.

### 4.4.1. Data Sources and Methods Used: The Case of ML

According to the analysis of the literature, the use of ML methods outweighs the use of DL, and very few works employ both types of methods (see Figure 9). Depending on the data sources, this analysis remains valid for data coming from sensors or hybrid sources. However, the exclusive use of ML methods is often seen for data coming from surveys, which often have significantly simpler structures to learn.

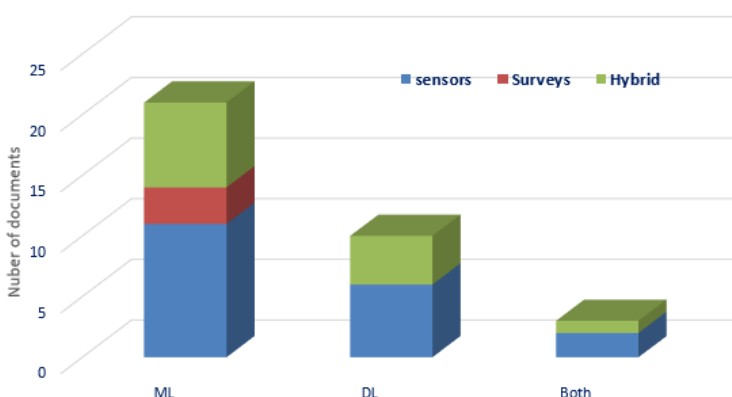

**Figure 9.** Use of ML depending on the data sources involved.

Our analysis supports our suggestion that the selection of ML methods depends on the used data source. More precisely, Figure 10 shows the proportions of the used method depending on the data sources, with ensemble methods in the lead, which are considered more robust than individual learning methods.

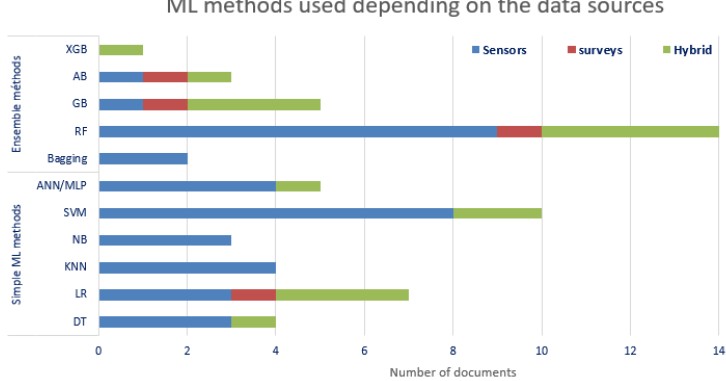

**Figure 10.** ML methods used depending on the data sources involved.

The analysis reveals that among these ML methods, the random forest (RF) algorithm is not only the most widely used, but also able to achieve the best performance in most use cases regardless of the data source. Next comes the support vector machine (SVM) algorithm, which is widely used for hybrid or sensor data, although it can be adapted to other data sources similarly effectively. Then come the logistic regression (LR), gradient boosting (GB), artificial neural network or multiple layer perceptron (ANN/MLP), decision trees (DT), k-nearest neighbor (KNN), naive Bayes (NB), and adaptive boosting (AB) algorithms. Algorithms such as XGboost (XGB) and bagging, however, are used relatively less frequently.

### 4.4.2. Data Sources and Methods Used: The Case of DL

For DL methods and the trends of their usage depending on data sources, the leading methods are based on deep neural networks (DNN) or convolutional neural networks (CNN) (Figure 11). They are most often effective for image processing, such as satellite images and remote sensing data. Then there is a strong prevalence of the recurrence learning methods, such as recurrent neural network (RNN) or long short-term memory (LSTM) that allow time series processing. Auto-encoders (AE) and generative adversarial neural networks (GAN) are also used for predictive regression and the generation of new artificial architectural forms [12]. Finally, other methods, including transfer learning (TL) and reinforcement learning (RL), are used relatively less frequently.

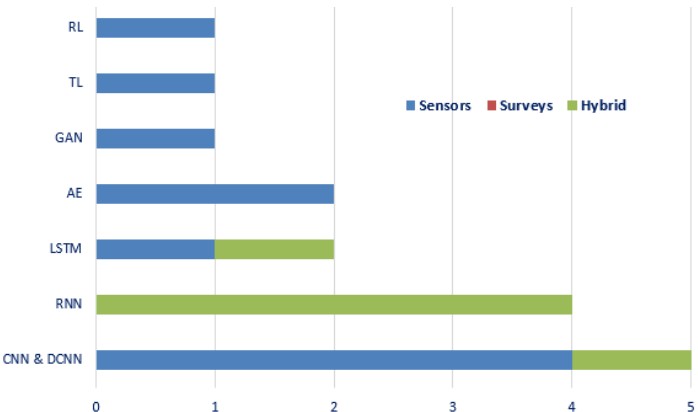

**Figure 11.** DL methods used depending on the data sources involved.

### 4.5. Urban Planning Issues According to the Urban Data Source

In Sections 2, 4.2 and 4.4, we discuss the structures, urban data sources, and trends on the selection of the ML method depending on the used data source. Now, let us discuss whether there are trends related to the use of data sources to address certain urban-planning problems. Our analysis resulted in the identification of seven urban planning issues, where some trends related to the selection of the data source can be observed. Figure 12 illustrates the urban-planning problems addressed by the data sources.

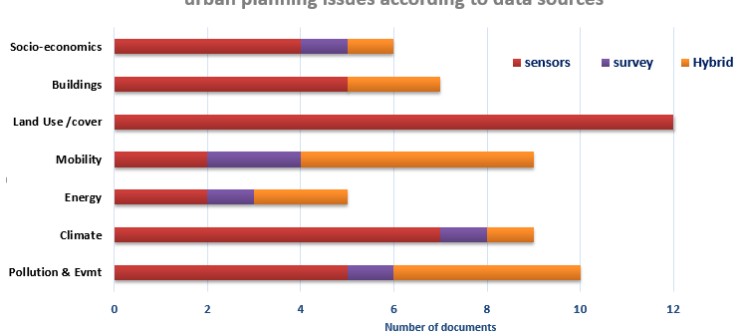

**Figure 12.** Urban-planning issues by urban data source.

When examining the type of urban planning problem considered with the reference to the data source used, i.e., in order to identify such relationship or dependence, we find that the land use/cover issue, which is assimilated to all classical urban planning, is the most addressed urban planning issue, which in most cases is solved using ML methods and only with the help of sensor data, more precisely, satellite data (see Figure 12). Key metrics or indicators targeted by this issue include land use [3], land cover [2], urban growth [1], and land values [47].

Then comes the issues of pollution and environment, climate and mobility, for which different data sources can be used. Pollution and environment, and mobility are two categories, where the ratio of studies using surveys is the highest compared to the identified issues.

They are followed by issues related to buildings/constructions, for which the data gathered from sensors and hybrid data sources are used with no studies found using survey data, making this issue, together with land use/land coverage, the only issues where surveys alone are not used. This allows us to speculate that this form of data sourcing is not sufficiently appropriate for this issue and should be avoided in newly launched studies, or at least its appropriateness should be extensively inspected, having a higher risk of project failure if it is used as the only method.

The list is concluded with the socioeconomic and energetic urban/energy problems, for which three data sources are used, although sensor data sources dominate with less expressed dominance of such for energy-related urban issues. For these various problems, we can refer to [16] for more details on related indicators.

## 5. Discussion and Conclusions

Today, there is a high importance of making it possible to integrate automated decision support into the planning process. However, the current body of knowledge demonstrates that many researchers and practitioners face various challenges, one of the most serious being related to the selection of urban data [16]. In this study, elements associated with urban data, i.e., their type, sources and structures, for better use in training ML methods were discussed.

We discussed the structuring of urban data, facilitated by the advent of GIS, which is more appropriate for storage, future accessibility, analysis, and processing, and, above all, for better analysis using ML methods. Urban data vectorization has improved the analysis and processing of initially complex spatial data to design and develop new applications useful for urban planning [1,23]. In addition, attribute data tables make it easier to analyze and visualize indicators, although often on a small scale, for greater precision and less complexity.

The urban data structured in these two major categories enabled the integration of new tools (ARGIC, QGIS, GEE, Python, etc.) and new intelligent applications based on ML methods. These applications are at the heart of solving the complex problems of the urban-planning system, as evidenced by the exponentially growing number of scientific contributions published in recent years. AI integration is seen as an asset to optimize urban planning and identify and optimize indicators of sustainability, resilience, socioeconomic inclusion, and overall development [16].

Our research has shown that the data used for this purpose mainly come from detection and are becoming increasingly complex. This complexity is marked by the increased use of intelligent ML-based models to support urban-planning decisions, including complex DSS.

Supervised methods, such as SVM and RF, are the most commonly used, regardless of the data type with the particularity of being explainable. They are followed by using unsupervised learning methods alone or with a supervised method afterwards to process unstructured data modeling. The use of deep learning methods is increasing to improve the efficiency of planning and real-time monitoring (e.g., of area), especially to solve issues, such as land use/coverage from vector (sized) satellite images. However, these complex models usually suffer from the explainability problem because they are of the "black box" type with an almost non-transparent decision-making process. We also provided insight into what data-sourcing methods are found to be the most appropriate for a particular urban planning issue, considering the current body of knowledge.

This paper does not address how or where the data are stored to avoid biasing the work with details not related to the data source or ML forming the core of this study. It also does not include data sources related to other processing techniques or issues other than urban planning, although they are quasi-similar, and the results can be extrapolated

to other domains of applications. We also covered only the Scopus database. While it is considered the most comprehensive overview of the world's research findings, and we believe that the results covering other databases would be the same since we were interested in the aggregated results and determination of the patterns, it can be still worthwhile to consider conducting this analysis covering more databases, e.g., including the Web of Science database.

We believe that this study will serve as a guide or reference for more active participation in addressing and solving urban-planning problems, constituting a knowledge base for selecting data sources and ML methods, including the reference to the urban planning issue to be addressed, that will capture and solve specific issues more accurately. In other words, the results of the study should be of interest not only to researchers and theoreticians, but also to practitioners; we believe they can be of particular interest to enthusiasts such as participants of marathons, hackathons and CityLab initiatives, letting them capture quicker what type of data and data source are the most appropriate to successfully implement the desired idea, thereby contributing to sustainable urban planning and development.

**Author Contributions:** Writing—original draft, S.C.K.T., H.T. and R.A.; Writing—review & editing, E.B.D. and A.N.; Supervision & administration, J.C.; Funding, J.C. and A.N. All authors have read and agreed to the published version of the manuscript.

**Funding:** This research has been co-funded by European Social Fund via the IT Academy program and the APC was funded by CEAT-ENAC-EPFL.

**Institutional Review Board Statement:** The study did not require ethical approval.

**Informed Consent Statement:** Not applicable.

**Data Availability Statement:** Not applicable.

**Conflicts of Interest:** The authors declare no conflict of interest.

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
