# Peer review of "Identifying and Classifying Urban Data Sources for Machine Learning-Based Sustainable Urban Planning and Decision Support Systems Development"

_data, 2018_

Round 1

Reviewer 1 Report

Congratulations to the authors for the great work done! My only suggestions/questions are as follows:

- Why did you only use the SCOPUS database? And WEB OF SCIENCE (WOS) AND SCIENCE DIRECT? I believe that if you are doing a systematic literature review, more than one database needs to be analyzed.

- Data mining could not be used as a keyword?

- It is necessary to separate the paragraphs better, it has very long paragraphs, and this makes it difficult to read.

In summary, the article is very good, congratulations!

Author Response

General Comments: Congratulations to the authors for the great work done!   Response to general comment: Thank you for your high evaluation! We are also grateful for your comments, which helped us improve the paper’s quality!
Comments # 1: My only suggestions/questions are as follows: Why did you only use the SCOPUS database? And WEB OF SCIENCE (WOS) AND SCIENCE DIRECT? I believe that if you are doing a systematic literature review, more than one database needs to be analyzed.
Response # 1: Thank you for this comment! The main search database was the scopus database, which has shown its reputation and popularity in several works on the theme of our paper. However, the list of papers selected was reinforced by the reference citation matching for the other databases you mentioned. This choice was also motivated by the access rights to all the documents in this database that our university subscription gives us.
Comments # 2: Data mining could not be used as a keyword?
Response # 2: Thank you for your suggestion, we now added ”text mining”, which corresponds to the technique we used, to the list of keywords.
Comments # 3: It is necessary to separate the paragraphs better, it has very long paragraphs, and this makes it difficult to read.   Response # 3: Thank you for this suggestion. Indeed, some paragraphs are too long. We reviewed the text and made sure that the text was split wisely into paragraphs. Now, we hope it is easier to read.
Comments # 4: In summary, the article is very good, congratulations!
Response # 4: Thank you very much! We are happy to hear this! Again, thank you for your efforts and useful suggestions!

  You will also find these responses on page 4 of the attached reviewer response document. 

Reviewer 2 Report

1)    The novelty and contribution of this paper are not very clear. Would you please add them either in abstract or in introduction?

2)    English is required to improve for example, in line 55, it is written as “In this paper, we discuss……” you could say the authors have discussed…..  

3)    Please review more relevant works and find the research gap from there.

4)    Performance/advantages comparison with existing related works would be good add at the end of chapter 4 to validate the capability of the proposed work.

5)    The authors could add Comparison of DL methods for urban form modelling.

6)    Figure 4 a and b should be more visible.

7)    ML for addressing urban sustainability issues can also be included.

8)    Please add strength, limitation and more impact/significance of this work in real life scenarios.

9)    potentials, challenges and future directions could be added

Author Response

Thank you for your comments, which helped us to improve the paper's quality. Your efforts are much appreciated. We have addressed as well as possible all the indicated issues in your review report and we believe that the revised manuscript can meet the journal publication requirements by convincing you as a kind reviewer. In this document, we propose detailed responses to all your comments and questions. Once again, all the revisions are highlighted in blue in the revised manuscript as well as the text extracts mentioned here in response to your comments (except those related to formatting).

Comments # 1: The novelty and contribution of this paper are not very clear. Would you please add them either in the abstract or in the introduction?
Response # 1: Thank you for this important note! We have extended an abstract to emphasize this point. This is also addressed in the Introduction.

Comments # 2: English is required to improve for example, in line 55, it is written as “In this paper, we discuss. . . . . . ” you could say the authors have discussed. . . ..
Response # 2: Thank you for the comment! We reread the paper in several iterations and tried our best to eliminate language issues. We also hope to have an opportunity to improve the language at a later stage, when the paper will be accepted by referring to a particular service provided by MDPI. For a particular comment, however, we would like to emphasize that this is not about language itself but rather about a writing style choice, i.e. about the use of active vs passive voice. We checked, and this journal, similar to many others, does not pose any requirements on this, and we are allowed to use active voice. I.e. this is a very popular debate, where those, who prefer to use an active voice, consider that it is not only acceptable but also effective in some cases, e.g., when the passive voice may introduce ambiguity (i.e. who exactly did it? some authors we ”forgot” to mention? or we?).

Comments # 3: Please review more relevant works and find the research gap from there.
Response # 3: Although the paper already refers to some works identifying the gap in the current knowledge, we would like to emphasize (and did this in the text) that it is not only of theoretical nature, and the need for this study mostly comes from the real world, i.e. today, many initiatives such as CityLabs, marathons, hackathons and other creative, but at the same time technical, events take place that brings together people with different backgrounds to solve a social issue, where this very fundamental understanding is needed. unfortunately, even those, who deal with these issues on a daily basis, often get an understanding of which data (data type), from which source should be used to tackle a particular issue, and, more importantly, how this should affect the selection of the technique to be used in dealing with these data. It takes a lot of time and effort to find an appropriate combination, while this is not even a main focus of the ”project”, where our study saves time, and effort and contributes to the digital literacy of those dealing with issues related to urban planning. Thank you for your comment, we now added this to the paper!

Comments # 4: Performance/advantages comparison with existing related works would be good add at the end of chapter 4 to validate the capability of the proposed work.

Response # 4: Considering that the objective of our study was to determine ”patterns” between urban planning issues, data types to be used to tackle them, the data sources for these data and the technique to be used, comparison of the performance with existing works is not possible (since the performance itself cannot be applied to SLR, for the best of our knowledge). The criterion we can use is the novelty (up-to-date) of the results, where our study presents the most up-to-date results given the choice of papers to study. Moreover, in light of the specifics of the objective we set, we were unable to find papers covering the same issue, so the comparison is even more challenging since this study is unique in this regard.

Comments # 5: The authors could add Comparison of DL methods for urban form modelling.
Response # 5: Thank you for this comment! It could be, indeed, an interesting point, however, in this study, we were more interested in ML (as suggested in the title), where what could lead to this comment most probably are some references to DL that we also found to be needed. However, it was not the focus of this study, but rather a complementary topic, keeping in mind that DL is rather a subset of ML. Considering that we were interested in the identification of those patterns between data types, data sources, techniques and urban planning issues, we needed a more substantial set of papers to derive these findings, while focusing on DL only would not allow getting generalizable results in the light of the limited number of studies on this topic (and used terminology, where DL tend to still be called ML).

Comments # 6: Figure 4a and b should be more visible.
Response # 6: We made them more visible, i.e. bigger in size. We also found it needed to do for other graphics.

Comments # 7: ML for addressing urban sustainability issues can also be included.
Response # 7: Thank you for the tip! We added this to the text!

Comments # 8: Please add strength, limitation and more impact/significance of this work in real-life scenarios.

Response # 8: Thank you for the note! We touch on this point in the Discussion and Conclusions.

Comments # 9: Potentials, challenges and future directions could be added
Response # 9: Thank you for the note! We touch on this point in the Discussion and Conclusions.

These responses can also be found on pages 5-7 of the attached reviewer response document. 

Round 2

Reviewer 2 Report

I would like to thank the authors for addressing some of my comments and explanation provided for those which have not been addressed. I have no other comments.